# Laparoscopic Function-Preserving Gastrectomy for Proximal Gastric Cancer or Esophagogastric Junction Cancer: A Narrative Review

**DOI:** 10.3390/cancers15010311

**Published:** 2023-01-03

**Authors:** Yosuke Kano, Manabu Ohashi, Souya Nunobe

**Affiliations:** 1Department of Gastroenterological Surgery, Gastroenterological Center, Cancer Institute Hospital, Japanese Foundation for Cancer Research, 3-8-31 Ariake, Koto-ku, Tokyo 135-8550, Japan; 2Division of Digestive and General Surgery, Niigata University Graduate School of Medical and Dental Sciences, Niigata 951-8510, Japan

**Keywords:** proximal gastric cancer, laparoscopic proximal gastrectomy, laparoscopic subtotal gastrectomy

## Abstract

**Simple Summary:**

Laparoscopic proximal gastrectomy (LPG) and laparoscopic distal gastrectomy with a small remnant stomach, namely laparoscopic subtotal gastrectomy (LsTG), are alternative function-preserving procedures for laparoscopic total gastrectomy (LTG) of early proximal gastric cancer. In this review, we summarize the status of reconstruction in LPG and the oncological and nutritional aspects of LsTG as a function-preserving gastrectomy for early proximal gastric or esophagogastric junction cancer. The overall incidences of anastomotic stenosis in esophagogastrostomy and esophagojejunostomy were comparable, although esophagogastrostomy using a circular stapler was associated with high rates of anastomotic stenosis and reflux esophagitis. Regarding post-operative nutritional status, esophagogastrostomy and esophagojejunostomy were also comparable. LsTG is also a feasible procedure for early proximal gastric cancer. However, it has indications for cancer in a limited area. The outcomes of LsTG are comparable to LPG, and LTG in oncological aspects, while it is superior to LTG in nutritional outcomes.

**Abstract:**

Function-preserving procedures to maintain postoperative quality of life are an important aspect of treatment for early gastric cancer. Laparoscopic proximal gastrectomy (LPG) and laparoscopic distal gastrectomy with a small remnant stomach, namely laparoscopic subtotal gastrectomy (LsTG), are alternative function-preserving procedures for laparoscopic total gastrectomy of early proximal gastric cancer. In LPG, esophagogastrostomy with techniques to prevent reflux and double-tract and jejunal interposition including esophagojejunostomy is usually chosen for reconstruction. The double-flap technique is currently a preferred reconstruction technique in Japan as an esophagogastrostomy approach to prevent reflux esophagitis. However, standardized reconstruction methods after LPG have not yet been established. In LsTG, preservation of the esophagogastric junction and the fundus prevents reflux and malnutrition, which may maintain quality of life. However, whether LsTG is an oncologically and nutritionally acceptable procedure compared with laparoscopic total gastrectomy or LPG is a concern. In this review, we summarize the status of reconstruction in LPG and the oncological and nutritional aspects of LsTG as a function-preserving gastrectomy for early proximal gastric or esophagogastric junction cancer.

## 1. Introduction

The survival of patients with early gastric cancer is now so favorable that the preservation of stomach functions to maintain postoperative quality of life (QOL) has become an important issue in the treatment of early gastric cancer [1,2]. Although function-preserving gastrectomy is not strictly defined, maintaining the stomach volume and structures that have specific functions, such as the cardia and the pylorus, is usually described as function-preserving gastrectomy. Laparoscopic total gastrectomy (LTG) is currently the standard procedure for early and even advanced proximal gastric cancer based on the results of some pivotal clinical trials [3,4]. Additionally, laparoscopic proximal gastrectomy (LPG) and laparoscopic distal gastrectomy with a small remnant stomach, namely subtotal gastrectomy (LsTG) [5], are adapted as function-preserving gastrectomy for such disease. Total gastrectomy (TG) may cause postoperative poor QOL because of malnutrition [6]. LPG and LsTG are performed as alternative procedures to maintain postoperative QOL by preserving the stomach volume and the pylorus or cardia.

LPG may be a suitable procedure for early proximal gastric cancer with regard to oncological aspects such as adequate lymph node dissection [7,8]. Furthermore, LPG has possible advantages regarding nutritional intake, including preserving the gastric volume and the pylorus, despite fewer gastric acid and hormone deficiencies. However, no standard reconstructive method for LPG has been established because few of these methods secure the balance between some clinical problems, such as anastomotic stenosis and gastroesophageal reflux.

Although the remnant stomach is extremely small, LsTG is basically a common procedure, with laparoscopic distal gastrectomy (LDG) performed for the transection of the stomach and reconstruction. Thus, it is easy to introduce this procedure instead of LPG. Furthermore, the postoperative outcomes of LsTG are predictable, based on many experiences of LDG. However, whether LsTG is an oncologically and nutritionally acceptable procedure for early proximal gastric cancer compared with LTG or LPG remains unclear.

In this review, we summarize the status of reconstruction in LPG, as well as the oncological and nutritional aspects of LsTG as function-preserving gastrectomy, for not only early proximal gastric cancer but also esophagogastric junction (EGJ) cancer, which can replace LTG.

## 2. Methods

Literature written in English, which were published from January 2000 to December 2021 and met the topic of this review, were collected using PubMed. The main keywords for the literature search were “gastric cancer”, “esophagogastric junction cancer”, “function-preserving gastrectomy”, laparoscopic proximal gastrectomy”, “laparoscopic subtotal gastrectomy”, “laparoscopic total gastrectomy”, “double-flap technique”, “esophagogastrostomy”, “double-tract reconstruction” and “jejunal interposition”. After searching, abstracts were first used to select proper literature and full texts of selected literature were additionally evaluated. We applied studies which included more than 10 patients undergoing LPG or LsTG to comparisons in some subjects and summarized the results in tables.

## 3. Literature Review

### 3.1. LPG

#### 3.1.1. Indication of LPG

The Japanese Gastric Cancer Treatment Guidelines (JGCTGs) state that LPG is an alternative procedure to LTG for cT1N0M0 tumors located in the upper third of the stomach regarding QOL and survival outcomes [9]. In LPG, D1+ lymphadenectomy was caried out including dissection of the lymph nodes at station numbers 1, 2 3a, 4sa, 4sb, 7, 8a, 9, and 11p [9]. Nationwide retrospective and prospective studies of lymph node metastasis in EGJ cancer in Japan showed an optimal lymphadenectomy region [10,11]. These studies demonstrated that the incidence of lymph node metastasis around the right gastric and right gastroepiploic artery area was zero to extremely low. Thus, proximal gastrectomy (PG) has a good indication not only for proximal gastric cancer but also for EGJ cancer. Furthermore, several studies also revealed that PG is not a limited procedure for early gastric cancer. According to the JGCTGs, the recommended surgery for upper third of stomach is TG with D2 nodal dissection for advanced disease. However, Ri et al. revealed that the frequencies of lymph node metastasis and therapeutic indices of suprapyloric nodes, infrapyloric nodes, and right greater curvature nodes along the right gastroepiploic artery are significantly low in advanced gastric cancer located in the upper third of the stomach [12]. Therefore, PG may be indicated for advanced gastric cancer in the upper third of the stomach considering the depth, size, and localization, as well as preoperative lymph node metastasis.

#### 3.1.2. Reconstruction Methods Following LPG

LPG can preserve more than half of the gastric volume and the pylorus, making it an ideal procedure as a function-preserving gastrectomy. However, LPG has the unavoidable problem of losing the cardia. The cardia prevents reflux in cooperation with the adjacent diaphragmatic crus and the phrenoesophageal ligament. After LPG, reconstructive devices to prevent reflux are required; in their absence, the contents of the remnant stomach are easily regurgitated, with specific symptoms such as heart burn, fore-chest pain, vomiting, and aspiration. Although many reconstruction methods for preventing reflux have been developed, a reconstruction method has not been definitively established. 

Esophagogastrostomy (EG) and esophagojejunostomy (EJ) are two major methods of reconstruction following LPG. EG is the simplest reconstruction method, but simple anastomotic EG does not avoid reflux. Thus, EG is usually accomplished with anti-reflux techniques. LPG with the double-flap technique (DFT) is one such technique and is currently a preferred reconstruction technique for LPG in Japan. However, double-tract (DT), jejunal interposition (JI), and jejunal pouch interposition are included in EJ after LPG. Among laparoscopic approaches, DT and JI are now common reconstruction methods including EJ.

#### 3.1.3. LPG-DFT

##### Surgical Procedures of LPG-DFT

The DFT was first reported by Kamikawa et al. in 2001 [13], and the detailed surgical procedure of EG with valvuloplasty by the DFT in LPG was described in recent reports [14,15,16]. Briefly, double flaps are created extracorporeally by dissecting between the submucosal and muscular layers on the anterior wall of the remnant stomach. After creating the seromuscular double flaps, the walls of the esophagus and gastric mucosa are sutured under laparoscopic view and an esophagogastrostomy is created. Finally, the hinged flaps are used to laparoscopically cover the anastomosis and lower esophagus.

##### Outcomes of LPG-DFT

Articles describing LPG-DFT are summarized in Table 1. The incidences of anastomotic stenosis, leakage, and reflux esophagitis were 0–29.1%, 0–7.7%, and 0–10.5%, respectively [14,15,16,17,18,19,20,21,22,23,24]. Furthermore, bodyweight loss (BWL), which may represent a postoperative nutritional outcome, was 8.5–15% [16,17,18,19,20,24]. Kuroda et al. reported the incidence of stenosis in LPG-DFT as 15%, but 5% in open PG with the DFT [14]. Furthermore, Shibasaki et al. reported that the incidence of stenosis was 25% in robot-assisted LPG-DFT [20]. Despite the low incidence of reflux esophagitis and leakage, the high occurrence of stenosis is an important problem of LPG-DFT. Several articles reported the risk of stenosis in LPG-DFT, and Shibasaki et al. presented the negative relationship between stenosis and the total number of stitches [20]. When performing LPG-DFT, an excessive number of stitches should be avoided because of the possibility of stenosis. The incidence of stenosis in LPG-DFT was higher than that in open PG-DFT and may be due to an excessive number of stitches under a magnified visual field of the laparoscopic view, which can lead to ischemia of the anastomosis. Furthermore, many surgeons adopt a continuous suture with a barbed string in LPG-DFT, which is often associated with stenosis. In robotic approaches, the lack of tactile feedback may lead to excessive tightening of stitches. Regarding other aspects, Shoji et al. reported a multivariate analysis that revealed that an esophageal diameter of <18 mm on pre-operative computed tomography images and the presence of short-term complications were independent risk factors for stenosis [25]. Muraoka et al. reported that the incidence of stenosis decreased from 50.0% to 8.3% after adopting intraoperative gastroendoscopy [15]. Considering these results, solutions for stenosis in LPG-DFT may include avoiding excessive stitches, a narrow esophagus, and postoperative complications, as well as using a gastroendoscope as a stent.

Another problem of LPG-DFT is the prolonged surgery time resulting from complex intracorporeal anastomotic procedures. The median or mean surgical times of LPG-DFT were more than 6 h in five of the nine published articles [15,16,18,20,21]. To solve such an essential problem, Omori et al. reported LPG-DFT using a stapler, which took a significantly shorter surgical time than conventional hand sewing anastomosis, although the other surgical outcomes such as reflux esophagitis were comparable [24].

##### LPG-DFT for EGJ Cancer

Although the DFT is a very effective reconstruction method for reflux esophagitis, it is controversial as to whether the DFT is a suitable reconstruction of LPG plus lower esophagectomy for EGJ cancer, which requires mediastinal or intrathoracic anastomosis. Mediastinal anastomosis is very complicated procedure in a limited surgical field, and negative pressure of the intrathoracic cavity may increase the risk of reflux esophagitis. In fact, Kuroda et al. reported that the incidence of reflux esophagitis was 18.2% for grade B or higher in patients whose DFT anastomosis was located in the mediastinum or intrathoracic cavity, and the anastomotic site in the mediastinum or intrathorax was one of the independent risk factors for reflux esophagitis [14]. However, Omori et al. showed that the incidence of reflux esophagitis was 6.9% after LPG plus lower esophagectomy with the DFT using a linear stapler for Siewert type II EGJ cancer [24]. Some modifications of the DFT may be necessary for performing effective DFT in the mediastinum or intrathoracic space.

#### 3.1.4. LPG-non-DFT

LPG-non-DFT using a circular stapler Table 2 summarizes a literature review of LPG-non-DFT. Most LPG-non-DFT is performed using a circular stapler. EG using a circular stapler is well known to have a high risk of reflux esophagitis in open PG [26,27]. Naturally, some types of techniques to prevent reflux esophagitis have been designed in LPG-non-DFT using a circular stapler [22,28,29,30,31]. However, the incidence of reflux esophagitis was still high, ranging 3.8–31.3% [22,28,29,30,31]. In addition, the incidence of anastomotic stenosis in this procedure ranged 13–27.5% [22,28,29,30,31]. In LPG, EG using a circular stapler may be not suitable for both stenosis and reflux esophagitis, similar to open PG. In LPG-non-DFT using a circular staple, the median or mean surgery times were less than 6 h except in one report [22,29,30,32]. BWL was 10.5–15% in the postoperative period [22,29,30,32].

#### 3.1.5. LPG-non-DFT Using a Linear Stapler

Ahn et al. reported that the incidence of stenosis was significantly higher in an end-to-end EG with a circular stapler than in a side-to-side EG with a liner stapler (46.2% vs. 0%, *p* < 0.001) [33]. Yamashita also reported that side overlap EG using a linear stapler with fundoplication, namely the side overlap with fundoplication by Yamashita (SOFY) method, was effective to avoid stenosis, leakage, and reflux esophagitis in comparison to EG with a circular stapler [32,34]. In the SOFY method, the left side of the esophageal wall is anastomosed with the anterior gastric wall using linear stapler and the right side of the esophageal wall is stuck to the gastric wall, which causes the preserved dorsal esophageal wall to be pressed and flattened into a valvate shape by pressure from the artificial fundus to form the reflux prevention mechanism [32]. Anastomosis using a linear stapler may be a more suitable technique for laparoscopic procedures than that using a circular stapler and is easier than that with an intracorporeal hand sewing suture [36,37]. Hence, anastomosis using a linear stapler that can prevent stenosis and reflux will be a common method for LPG-EG if favorable long-term surgical results are obtained. In LPG-non-DFT using a linear stapler, the median or mean surgery times were less than 6 h [32,34,35]. BWL was 7.4% in the postoperative period [32].

#### 3.1.6. LPG-DT and JI

Table 3 details a literature review of LPG-DT and JI. The incidences of anastomotic stenosis, leakage, and reflux esophagitis in the DT were reported to be 0–21.4%, 0–10%, and 6.7–25%, respectively [7,23,30,35,38,39,40,41,42,43]. Those in the JI were 0–20%, 0–9.5%, and 0–10%, respectively [28,39,40,44,45]. The incidence of stenosis in EJ was observed at a certain rate for a circular stapler but was 0% for a linear stapler except for one report [7,23,39,40,41,42,43,44,45], while the incidence of stenosis in EJ of LTG with a circular stapler was reported as 7.1–7.7% [46,47]. Thus, EJ with a circular stapler has a risk of stenosis in both LTG and LPG. Recently, EJ was mainly performed with a linear stapler as overlapping or functional end-to-end anastomotic methods. The incidence of stenosis in EJ of LTG using a liner stapler is significantly lower than that using a circular stapler [46,47]. Therefore, the use of a linear stapler in the DT or JI may improve the incidence of stenosis.

In both the DT and JI, the small intestine is cut and lifted to interpose between the esophagus and the stomach to prevent reflux esophagitis. Such usage of the small intestine can induce several issues. One is small bowel obstruction due to adhesion and another is difficulty in performing endoscopic surveillance of the remnant stomach. The incidences of small bowel obstruction and impossible surveillance were reported to be 9.4–20.0% and 7–50%, respectively [28,35,39,41,46,48,49]. In PG, 5.0–9.1% patients experience remnant stomach cancer or newly arisen cancer [50,51]. Hence, the simplicity of postoperative surveillance makes it an important factor in choosing the method of reconstruction following LPG.

Although LPG-DT and JI require multiple anastomoses, the mean or median surgical time was within 6 h in all reports [7,30,35,38,40,41,42,43,44,45]. BWL in LPG-DT and JI was 9.6–12.4% and 8.9%, respectively [7,30,38,39,42,43]. In LPG-DT, there are some patients in whom ingested foods do not pass through the remnant stomach, but the values of BWL were reported to be comparable to the other LPG reconstruction methods.

**Table 3 cancers-15-00311-t003:** Summary of LPG-DT and JI literature.

Author	n	Approach	EJ Anastomotic Method	Time, min	Blood Loss, mL	Anastomotic Stenosis	Anastomotic Leakage	Reflux Esophagitis * (Month after Surgery)	BWL (Month after Surgery)
**Double-tract**									
Jung [7]	92	Laparoscopic	Circular	198.3 ^a^	84.7 ^a^	EJ: 3.3%	2.2%	NA	10.22% ^a^ (12 M) 9.36% ^a^ (24 M)
Aburatani [30]	19	Laparoscopic	Circular	325.7 ^a^	131.4 ^a^	0%	0%	10.5% (12 M)	12.4% ^a^ (12 M)
Sakuramoto [35]	10	Laparoscopic	Circular	235 ^b^	60 ^b^	10% ^c^	0%	25% (12 M)	NA
Ahn [38]	43	Laparoscopic	Circular	180.7 ^a^	120.4 ^a^	4.65% ^c^	NA	NA	5.9% ^a^ (6 M)
Nomura [39]	10	Laparoscopic	Circular	NA	NA	EJ: 10%	NA	10% ^h^	NA
Nomura [40]	15	Laparoscopic	Circular	352.5 ^a^	90.5 ^a^	EJ: 6.7%	0%	6.7% ^d,h^	11% ^a^ (12 M)
Saze [23]	14	Laparoscopic	Linear	NA	NA	21.4%	0%	21.4% ^c^	NA
Cho [41]	38	Laparoscopic	Linear	217.7 ^a^	100.2 ^a^	0%	2.6%	NA	NA
Sugiyama [42]	10	Laparoscopic	Linear	341.9 ^a^	179.8 ^a^	0%	10%	NA	9.6% ^a^ (12 M)
Xiao [43]	46	Laparoscopic	Linear	258 ^a^	NA	0%	2.2%	NA	7.0% ^a^ (6 M)
Park [52]	34	Laparoscopic	Linear	212.9 ^a^	30 ^b^	NA	NA	NA	NA
**Jejunal interposition**									
Yasuda [28]	21	Laparoscopic (*n* = 5) Open (*n* = 16)	Circular	268.8 ^a^	307.4 ^a^	14.3% ^c^ (early ^f^) 10% ^c^ (late ^g^)	9.5%	0% (12 M)	NA
Nomura [39]	10	Laparoscopic	Circular	NA	NA	EJ: 20%	NA	10% ^h^	NA
Nomura [40]	15	Laparoscopic	Circular	322.5 ^a^	46.8 ^a^	EJ: 6.7%	0%	6.7% ^d,h^	8.9% ^a^ (12 M)
Kinoshita [44]	90	Laparoscopic (*n* = 22)	Circular	233 ^b^	20 ^b^	EJ: 9.1%	9.1%	1.1% ^e,h^	NA
		Open (*n* = 68)	Circular	201 ^b^	242 ^b^	EJ: 5.9%	7.4%	NA	NA
Takayama [45]	70	Laparoscopic (*n* = 32)	Circular	189 ^b^	30 ^b^	EJ: 3.1%	0%	4% (12 M)	NA
		Open (*n* = 38)	Circular	154 ^b^	180 ^b^	0%	0%	0% (12 M)	NA

LPG-DT and JI, laparoscopic proximal gastrectomy with double tract and jejunal interposition; BWL, body weight loss; EJ, esophagojejunostomy; NA, not available; M, months. * Reflux esophagitis is classified according to the Los Angeles classification. Values are Grade B or more. ^a^ Mean values. ^b^ Median values. ^c^ Data of anastomotic site with stenosis not available. ^d^ Value including Grade A or more. ^e^ Grade not available. ^f^ Anastomotic stenosis occurred within 1 month after surgery. ^g^ Anastomotic stenosis occurred after 1 month after surgery. ^h^ Timing of evaluation not available.

### 3.2. LsTG

#### 3.2.1. Specific Features of LsTG

In the early days of laparoscopic procedures, when LPG and LTG surgical results were inadequate, LsTG was first reported in 2011 by Jiang et al. as another procedure for early proximal gastric cancer [5]. The surgical procedure for this approach was described in previous reports [5,18]. Although it is commonly the same procedure as that of conventional LDG, there is the occasional requirement for lymph node dissection along the splenic artery (around the posterior gastric artery) in addition to D1+ lymphadenectomy including dissection of the lymph nodes at station numbers 1, 3a, 3b, 4sb, 5, 6, 7, 8a, and 9, and securing an oral margin by intraoperative endoscopy with intraoperative frozen section analysis is conducted at a different point. In LsTG, securing an oral margin is the most technically essential point. Placement of marking clips and intraoperative endoscopy is effective in determining a gastric transection line for LsTG. Kawakatsu et al. showed that the success rate of achieving a negative surgical margin during the initial transection was 98.9% in patients who underwent preoperative placement of marking clips and intraoperative endoscopy [53]. However, in patients with a proximal tumor extremely close to the cardia or fornix who are eligible for LsTG, conventional marking with clip placement might be problematic because of the risk of wedging by a linear stapler. As a safe technique for securing an oral margin in such patients, Kamiya et al. established that endoscopic cautery marking involving the use of endoscopic forceps cauterization was effective in determining a gastric transection line [54].

#### 3.2.2. Indication of LsTG

Although LsTG is one procedure for proximal gastric cancer, the indication of LsTG has several limitations. Table 4 shows a literature review of LsTG. In four of the five articles, LsTG was performed for cT1N0 or Stage I disease. LsTG was usually performed in patients who fulfil the following criteria: first, early gastric cancer diagnosed as cT1N0; second, tumor located in or involving the upper third of the stomach; and third, the proximal boundary of the tumor is more than 3 cm from the EGJ. Although the new marking technique described above enables transection of the stomach closer to the cardia, disease that is located extremely close to the EGJ or in the fundus is not an indication for this procedure. Nakauchi et al. reported that the survival of LsTG for advanced gastric cancer was comparable to that of conventional LDG for advanced gastric cancer [55]. However, this is the only report regarding LsTG for advanced gastric cancer. Thus, whether the indication of LsTG for advanced gastric cancer is adequate remains unclear. Furthermore, there are still oncological and nutritional concerns in LsTG for early gastric cancer.

#### 3.2.3. Oncological Problems of LsTG

Regarding the oncological aspects, LsTG is essentially associated with an insufficient proximal margin because of its proximity to the cardia and the risk of inadequate lymph node dissection, especially the left cardial and left greater curvature nodes along the short gastric arteries, despite proximal gastric cancer. Kano et al. reported that LsTG was oncologically feasible for cT1N0M0 gastric cancer located in the upper gastric body because of the extremely low incidence of metastases at such lymph node stations and had 3-year overall survival and relapse-free survival rates equivalent to those of LPG and LTG [59]. However, the length of the proximal margin in LsTG was significantly shorter than those in LPG and LTG. Another aspect of the oncological problem of short proximal margin length is whether the length is associated with survival outcome, which has been controversial [26,60,61,62,63,64]. However, Hayami et al. revealed that shorter proximal margin lengths than the recommendations of the JGCTGs in early gastric cancer did not affect survival outcome [65].

#### 3.2.4. Nutritional Problems of LsTG

The remnant stomach after LsTG nearly consists of only the cardia and fornix. Whether such an extremely small proximal remnant stomach works effectively for maintaining postoperative nutrition and QOL is another issue of LsTG. Mean or median BWL after LsTG was approximately 10–12%, except for one report that reported 4–6% BWL [18,55,56,57,58]. Compared with LTG or LPG, BWL after LsTG was comparable to that in LPG [18,57], while it was significantly lower than that in LTG [55,56,57]. Furthermore, it is generally assumed that the grade of BWL after LsTG is higher compared with that of conventional LDG. Yasufuku et al. reported that although the difference in BWL between LsTG and conventional LDG was statistically significant, it was only approximately 2% and might not strongly influence the QOL of patients undergoing LsTG [58].

Regarding nutritional parameters at certain times after surgery, Kosuga et al. reported that serum total protein (TP) and albumin (Alb) levels in LsTG were significantly higher than those in LTG [56]. Furukawa et al. reported that LsTG resulted in better serum Alb and prognostic nutritional index levels than LPG, and hemoglobin (Hb) levels in LsTG were significantly higher than in LTG [57]. Yasufuku et al. reported that TP and Alb levels after LsTG were comparable to those in conventional LDG, but Hb levels in LsTG were significantly lower than those in conventional LDG [58]. However, Kano et al. showed that TP, Alb, and Hb levels at 1 year after surgery were comparable between LsTG and LPG-DFT, but Hb levels at 3 years after LsTG were significantly lower than those after LPG-DFT [18].

#### 3.2.5. Reflux Esophagitis after LsTG

LsTG confers a risk of reflux esophagitis compared with conventional LDG because of the issue of hiatal hernia, which is caused by the destruction of the normal structure around the EGJ in sufficient lymph node dissection. However, the incidence of reflux esophagitis after LsTG was reported to be 0–4% [18,56,57,58], which is feasible compared with that after LPG. In fact, a Japanese multi-center study recently revealed that (L)sTG was associated with better postgastrectomy symptoms including esophageal reflux and daily lives than (L)TG using the Postgastrectomy Syndrome Assessment Scale-45 [66].

### 3.3. LPG vs. LsTG

Although both LPG and LsTG are surgeries for cancer in the upper stomach, they are opposite-side procedures. In LPG, the upper stomach is removed and the middle to lower stomach is preserved. Conversely, in LsTG, the middle to lower stomach is completely removed. Thus, indications for both procedures are essentially different. However, the indications sometimes overlap. When a tumor is located in the upper gastric body, both procedures can be performed. In such a case, the surgeon must select which procedure to perform, LPG or LsTG. Table 5 presents the differences between the two procedures according to the current literature [7,14,15,16,17,18,19,20,21,22,23,24,29,30,32,33,34,35,36,37,38,39,40,41,42,43,44,45,55,56,57,58]. The oncological and nutritional outcomes were basically comparable in both procedures. Regarding the resection margin length and anemia as a long-term outcome, LPG was superior to LsTG, although the surgery time of LsTG was shorter than that of LPG.

## 4. Discussion

LPG is an oncologically feasible procedure for early gastric cancer in the upper third of the stomach and either early or advanced cancer in the EGJ. Furthermore, even advanced gastric cancer may be applicable if the tumor has some specific conditions. However, the most proper reconstructive method has not been established. In the studies described in this review, the overall incidences of anastomotic stenosis in EG and EJ were comparable, although EG using a circular stapler was associated with high rates of anastomotic stenosis and reflux esophagitis; however, those of reflux esophagitis were different. Those in the DFT and the SOFY method, both of which are EG, were up to 10%, in which there was little variation among institutions as shown in Table 1, while those in the others could be more than 20%. Regarding post-operative nutritional status, EG and EJ were also comparable in this review, although some reports revealed that EG was superior to EJ regarding BWL, subjective symptoms, and the risk of intestinal obstruction [23,28,31,67,68,69]. The DFT or EG using a linear stapler (the SOFY method) has enough anti-reflux mechanisms, a physiological fashion of the gastrointestinal tract, and a benefit for postoperative endoscopic surveillance, and is associated with relatively satisfactory outcomes. However, problems with the DFT remain to be solved including the high incidence of anastomotic stenosis and the 6 h plus surgery time. Stenosis may be decreased by considering the number or strength of stitches, and the time-consuming element will be solved by technical improvement and machine usage. From this point of view, the SOFY method is near-ideal because it has a low risk of stenosis and does not need as long a surgery time. However, the best way of reconstruction after LPG should be determined considering not only objective surgical outcomes but also patient-reported outcomes. Thus, this clinical question remains unmet and further studies are required.

LsTG is also a feasible procedure for early proximal gastric cancer. However, it has indications for cancer in a limited area. The outcomes of LsTG are comparable to conventional LDG, LPG, and LTG in oncological aspects, while it is superior to LTG in nutritional outcomes. The procedures of LsTG are basically the same as those of conventional LDG. Thus, the only technical problem of LsTG is maintenance of the proximal margin length and its negativity for cancer. Preoperative marking and intraoperative techniques to ensure complete resection of the tumor are the most important issues in LsTG. In transecting the stomach in LsTG, surgeons should especially pay attention to preventing positive margins. Furthermore, the cutting margin should be submitted to intraoperative frozen section analysis to ensure a pathologically negative proximal margin. When the margin is positive for cancer, surgeons should not hesitate to convert to LTG to obtain a negative proximal margin. If intraoperative frozen section analysis is not available, LsTG should not be conducted. Although the proximal margin is pathologically negative, the length is extremely short. Whether the short proximal margin length affects oncological outcomes is an important issue in performing LsTG. Thus, the association between the margin length and survival outcome should be elucidated.

Although the remnant stomach is extremely small in LsTG, the nutritional status of LsTG is comparable to LPG and conventional LDG, except for Hb level, and is better than that of LTG. One reason for such a nutritional status may be the preservation of the gastric fundus, which is the primary location of ghrelin secretion, a gut hormone known to increase appetite [70]. Even though the preserved stomach is extremely small, preservation of the fundus may be associated with maintained appetite in LsTG. Nonetheless, Hb levels in patients who have undergone LsTG decrease over time because of decreased iron uptake, iron storage, or decreased vitamin B12 concentration. Iron absorption occurs mainly when ingested food passes through the duodenum. However, food passes through the duodenum after LPG-DFT, but not after LsTG. Furthermore, almost all the parietal cells, which are necessary for the absorption of vitamin B12 in the terminal ileum, are resected in LsTG. Thus, this might be associated with the occurrence of vitamin B12 deficiency anemia after LsTG.

The low incidence of reflux esophagitis leads to maintenance of the postoperative QOL of patients who undergo LsTG. A small remnant stomach and the EGJ may work to a certain degree. Furthermore, there may be little acid secretion from an extremely small remnant stomach, and it cannot store food. Thus, food residue and gastric acid does not remain in the remnant stomach but immediately passes to the jejunum. Even if bile reflux occurs, the anti-reflux mechanism of the EGJ may function to some extent.

When disease is located in the upper gastric body where the indications for LPG and LsTG overlap, LPG is first recommended because it confers several oncological and nutritional advantages. However, LPG includes complicated procedures regardless of the reconstruction method selected by the surgeon and requires a longer surgery time. In contrast, LsTG is one type of LDG. The procedures are simple and common for surgeons, who have many experiences of LDG and patients who have undergone LDG. LsTG may be more suitable considering the safety, familiarity, reliability of procedures, and well-experienced postoperative management. Thus, if the proximal margin length can definitely be maintained or if surgeons are not familiar with LPG, LsTG may be the first consideration for such disease.

This review has several limitations. First, this is not a systematic review but a classical narrative one. Literature was not systematically searched for and that which met our topics was only collected. Inclusion and exclusion criteria of collected literature was not prospectively determined. Thus, this review potentially has a heavy selection bias, which is just a nature of narrative review. Second, a lot of literature collected for this review was published from east Asian countries. In Western countries, early gastric or EGJ cancer is not frequently found and few studies regarding function-preserving gastrectomy were published. Considering these facts, function-preserving gastrectomy may interest surgeons in the limited area. On the other hand, this review has a strength. LPG or LsTG for proximal gastric or EGJ cancer is selectable according to the slight differences of location and both procedures are sometimes applicable. Few studies have been reported considering these issues. Many surgeons can refer our review when they consider which procedure they should conduct.

## 5. Conclusions

In conclusion, function-preserving gastrectomy may benefit patients with early proximal gastric or even advanced EGJ cancer. The most urgent issue is to establish the best reconstruction approach after LPG, and well-designed clinical trials should be conducted in the future. LsTG has narrow indications and requires specific considerations for ensuring the proximal margin. However, the basic procedures are common to those of conventional LDG, which is most familiar to surgeons. LsTG easily realizes preservation of gastric function and provides patients with a favorable QOL. Laparoscopic function-preserving gastrectomy for proximal gastric cancer or EGJ cancer is supposed to benefit patients. However, surgeons should understand the features of each procedure, determine the proper indications, upgrade their own skills, and generate reliable evidence to provide more patients with the benefits of such surgery.

## Figures and Tables

**Table 1 cancers-15-00311-t001:** Summary of LPG-DFT literature.

Author	n	Approach	Time, min	Blood Loss, mL	Anastomotic Stenosis	Anastomotic Leakage	Reflux Esophagitis * (Month after Surgery)	BWL (Month after Surgery)
Kuroda [14]	33	Laparoscopic (*n* = 13)	342 ^b^	NA	15%	0%	0% (12 M)	NA
		Open (*n* = 20)	288 ^b^	NA	5%	0%	0% (12 M)	NA
Muraoka [15]	24	Laparoscopic	372 ^a^	108 ^a^	29.1%	4.2%	4.2% ^c^	NA
Hayami [16]	43	Laparoscopic	386.5 ^a^	75 ^a^	4.7%	0%	2.3% (12 M)	10–15% ^b^ (12 M)
Kuroda [17]	464	Laparoscopic (*n* = 84) Open (*n* = 380)	298 ^b^	240 ^b^	5.5% (LPG16.7%)	1.5%	6% (12 M)	11.3% ^b^ (12 M)
Kano [18]	51	Laparoscopic	404 ^b^	68 ^b^	8%	0%	2% (12 M)	10–12% ^b^ (12 M)
							4% (36 M)	10–12% ^b^ (36 M)
Tsumura [19]	16	Laparoscopic	280 ^b^	210 ^b^	5%	0%	NA	10.4% ^a^ (6 M) 9.8% ^a^ (12 M)
Shibasaki [20]	12	Robotic	406 ^b^	31 ^b^	25%	0%	8.3% (6 M)	8.5% ^b^ (6 M)
Saeki [21]	13	Laparoscopic	389 ^a^	110 ^a^	0%	7.7%	0% (12 M)	NA
Hosoda [22]	40	Laparoscopic	353 ^b^	65 ^b^	18%	2.5%	8.3% ^c^	NA
Saze [23]	36	Laparoscopic (*n* = 20) Robotic (*n* = 13) Open (*n* = 3)	NA	NA	8.3%	2.8%	0% ^c^	NA
Omori [24]	59	Laparoscopic	316 ^b^	22.5 ^b^	5.1%	1.7%	10.5% (12 M)	11.6% (12 M)

LPG-DFT, laparoscopic proximal gastrectomy with double flap technique; BWL, body weight loss; M, months; NA, not available. * Reflux esophagitis classified according to the Los Angeles classification. Values are Grade B or more. ^a^ Mean values. ^b^ Median values. ^c^ Timing of evaluation not available.

**Table 2 cancers-15-00311-t002:** Summary of LPG-non-DFT literature.

Author	n	Approach	Anastomotic Method	Anti-Reflux Procedure	Time, min	Blood Loss, mL	Anastomotic Stenosis	Anastomotic Leakage	Reflux Esophagitis * (Month after Surgery)	BWL (Month after Surgery)
Hosoda [22]	40	Laparoscopic	Circular	Performed	280 ^b^	70 ^b^	27.5%	5%	5% (12 M)	12.8% ^a^ (12 M) 12.9% ^a^ (24 M)
Yasuda [28]	25	Laparoscopic (*n* = 20) Open (*n* = 5)	Circular	Performed	286.4 ^a^	294.2 ^a^	21.7%	0%	13.6% (12 M)	NA
Kosuga [29]	25	Laparoscopic	Circular	Performed	373 ^b^	40 ^b^	16%	0%	9.1% (12 M)	12.2% ^a^ (12 M) 10.5% ^a^ (24 M)
Aburatani [30]	22	Laparoscopic	Circular	Performed	290.3 ^a^	132.0 ^a^	27.2%	0%	22.7% (12 M)	12.6% ^a^ (6 M) 12.2% ^a^ (12 M)
Toyomasu [31]	84	Laparoscopic (*n* = 69) Open (*n* = 15)	Circular	Performed	204.2 ^a^	208.9 ^a^	13%	2.5%	3.8% (12 M)	15–20% ^a^ (12 M) 5–10% ^a^ (60 M)
Yamashita [32]	30	Laparoscopic	Circular (*n* = 16)	NA	337 ^a^	61 ^a^	18.6%	12.5%	31.3% ^c^	15.0% ^a^ (12 M)
			Linear (*n* = 14)	Performed	330 ^a^	17 ^a^	0%	0%	10% ^c^	7.4% ^a^ (12 M)
Ahn [33]	50	Laparoscopic	Circular (*n* = 13)	Not performed	216.3 ^a^	115.8 ^a^	46.2%	NA	NA	NA
			Linear (*n* = 37)	Performed			0%			
Yamashita [34]	36	Laparoscopic	Linear	Performed	302 ^b^	10 ^b^	2.8%	0%	10.7% ^c^	NA
Sakuramoto [35]	26	Laparoscopic	Linear	Performed	292 ^b^	90 ^b^	0%	7.7%	20% (12 M)	NA
Nishigori [36]	20	Laparoscopic	hand-sewn	Performed	300 ^b^	30 ^b^	25%	5%	5% ^c^	10.7% ^b^ (12 M)
Komatsu [37]	23	Laparoscopic	hand-sewn	Performed	325 ^b^	64 ^b^	4.30%	0%	0% ^c^	7.5% ^a^ (6 M)

LPG-non-DFT, laparoscopic proximal gastrectomy with non-double flap technique; BWL, body weight loss; M, months. * Reflux esophagitis classified according to the Los Angeles classification. Values are Grade B or more. ^a^ Mean values. ^b^ Median values. ^c^ Timing of evaluation not available.

**Table 4 cancers-15-00311-t004:** Summary of LsTG literature.

Author	n	Stage	Time, min	Anastomotic Stenosis	Anastomotic Leakage	Reflux Esophagitis *	BWL	Comparison of Nutritional Value between Procedures ^c^				
								BW	TP	Alb	Hb	PNI
Kano [18]	110	T1N0	289 ^b^	2.7%	0%	0%	10–11% ^b^	LsTG = LPG	LsTG = LPG	LsTG = LPG	LsTG < LPG	NA
Nakauchi [55]	27	≥Stage IB	333 ^b^	0%	0%	NA	12.7% ^b^	LsTG > LTG LsTG = LDG	NA	NA	NA	NA
Kosuga [56]	57	T1N0	289.3 ^a^	3.5%	0%	0%	10.2% ^a^	LsTG > LTG	LsTG > LTG	LsTG > LTG	NA	NA
Furukawa [57]	38	Stage I	274 ^b^	0%	3%	4%	4–6% ^b^	LsTG = LPG LsTG > LTG	NA	LsTG > LPG LsTG = LTG	LsTG > LTG LsTG = LPG	LsTG = LTG LsTG > LPG
Yasufuku [58]	73	Stage I	268 ^b^	NA	0%	0%	10.4% ^b^	LsTG < LDG	LsTG = LDG	LsTG = LDG	LsTG < LDG	NA

LsTG, laparoscopic subtotal gastrectomy; BWL, body weight loss; BW, body weight; TP, total protein; Alb, albumin; Hb, hemoglobin; PNI, prognostic nutritional index; LPG, laparoscopic proximal gastrectomy; LTG, laparoscopic total gastrectomy; NA, not available. * Reflux esophagitis is classified according to the Los Angeles classification. Values are Grade B or more. ^a^ Mean values. ^b^ Median values. ^c^ Mean sign of inequality.

**Table 5 cancers-15-00311-t005:** Comparative outcomes of LPG and LsTG.

	Surgical Outcome [7,14,15,16,17,18,19,20,21,22,23,24,29,30,32,33,34,35,36,37,38,39,40,41,42,43,44,45,55,56,57,58]				Oncological Outcome [59]		Nutritional Outcome [18,57]			
Procedure	Time, min	Anastomotic Stenosis	Anastomotic Leakage	Reflux Esophagitis	PM Length	OS	TP	Alb	Hb	BWL
LPG	189–389	0–46.2%	0–12.5%	0–31.3%	LsTG < LPG	LsTG = LPG	LsTG = PG	LsTG ≥ LPG	LsTG ≤ PG	LsTG = PG
LsTG	274–333	0–3.5%	0–3%	0–4%						

LPG, laparoscopic proximal gastrectomy; LsTG, laparoscopic subtotal gastrectomy; PM, proximal margin; OS, overall survival; TP, total protein; Alb, albumin; Hb, hemoglobin; BWL, body weight loss.

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
