# Peer review of "Laparoscopic Function-Preserving Gastrectomy for Proximal Gastric Cancer or Esophagogastric Junction Cancer: A Narrative Review"

_cancers, 2023, doi:10.3390/cancers15010311_

Round 1

Reviewer 1 Report

We thank the authors for submitting the present interesting review. Function-preserving gastrectomy for early gastric cancer is a topic gaining in popularity and it would be worth identifying eligible patient subgroups to apply alternative surgical techniques to both ensure oncological outcomes and improve quality of life.

In their effort to fulfil this knowledge gap, the authors presented an interesting report in a clear and understandable language, with an abstract mostly representing an accurate summary of the key- questions and outcomes. The Introduction Section attracts the reader’s attention, gives an idea of the topic and encompasses the purpose of the present report, which is to summarize the status of reconstruction in LPG, as well as the oncological and nutritional aspects of LsTG as function-preserving gastrectomy, for both early proximal gastric cancer and esophagogastric junction cancer, which could replace LTG.

However, the manuscript lacks the sections: Methods/patients and Results:

As the present manuscript is drawing its conclusions by reviewing the literature with regard to function-preserving types of gastrectomy for early gastric cancer, the readers would expect to see a clear, accurate and thorough methodology behind the literature search performed to extract the appropriate literature data. Specific criteria, which determined which studies were included or excluded in the final analysis, should also be stated in the Methods section.

For example: A paradigm of literature search with keywords: ‘ function preserving gastrectomy’ yields results as below:

1.      Performing robot-assisted pylorus and vagus nerve-preserving gastrectomy for early gastric cancer: A case series of initial experience. Zhang C, Wei MH, Cao L, Liu YF, Liang P, Hu X. World J Gastrointest Surg. 2022 Oct 27;14(10):1107-1119. doi: 10.4240/wjgs.v14.i10.1107. PMID: 36386400

2.      Potential Applicability of Local Resection With Prophylactic Left Gastric Artery Basin Dissection for Early-Stage Gastric Cancer in the Upper Third of the Stomach. Akashi Y, Ogawa K, Hisakura K, Enomoto T, Ohara Y, Owada Y, Hashimoto S, Takahashi K, Shimomura O, Doi M, Miyazaki Y, Furuya K, Moue S, Oda T. J Gastric Cancer. 2022 Jul;22(3):184-196. doi: 10.5230/jgc.2022.22.e17. PMID: 35938365

3.      Short-Term Outcomes of Laparoscopic Proximal Gastrectomy With Double-Tract Reconstruction Versus Laparoscopic Total Gastrectomy for Upper Early Gastric Cancer: A KLASS 05 Randomized Clinical Trial. Hwang SH, Park DJ, Kim HH, Hyung WJ, Hur H, Yang HK, Lee HJ, Kim HI, Kong SH, Kim YW, Lee HH, Kim BS, Park YK, Lee YJ, Ahn SH, Lee IS, Suh YS, Park JH, Ahn S, Han SU. J Gastric Cancer. 2022 Apr;22(2):94-106. doi: 10.5230/jgc.2022.22.e8. PMID: 35534447

4.      [Pylorus-preserving gastrectomy and segmental gastrectomy: discrimination of concepts and surgical procedures]. Xia MJ, Wang Q. Zhonghua Wei Chang Wai Ke Za Zhi. 2021 May 25;24(5):454-457. doi: 10.3760/cma.j.issn.441530-20210301-00087. PMID: 34000777

5.      Laparoscopic-assisted versus open proximal gastrectomy with double-tract reconstruction for Siewert type II-III adenocarcinomas of esophago-gastric junction: a retrospective observational study of short-term outcomes. Zhang B, Liu X, Ma F, Peng L, Lu S, Zhang Y, Ma Q, Ji S, Zhang Z, Chai J, Hua Y, Wang H, Li Q, Luo S, Chen X. J Gastrointest Oncol. 2021 Apr;12(2):249-258. doi: 10.21037/jgo-21-165. PMID: 34012623

6.      Assessment of Lymph Node Metastasis in Patients With Gastric Cancer to Identify Those Suitable for Middle Segmental Gastrectomy. Khalayleh H, Kim YW, Yoon HM, Ryu KW. JAMA Netw Open. 2021 Mar 1;4(3):e211840. doi: 10.1001/jamanetworkopen.2021.1840. PMID: 33729506

7.      Double-Tract Reconstruction Designed to Allow More Food Flow to the Remnant Stomach After Laparoscopic Proximal Gastrectomy. Fujimoto D, Taniguchi K, Kobayashi H. World J Surg. 2020 Aug;44(8):2728-2735. doi: 10.1007/s00268-020-05496-0. PMID: 32236727

8.      Feasibility of totally laparoscopic pylorus-preserving gastrectomy with intracorporeal gastro-gastrostomy for early gastric cancer: a retrospective cohort study. Akiyama Y, Sasaki A, Iwaya T, Fujisawa R, Sasaki N, Nikai H, Endo F, Baba S, Hasegawa Y, Kimura T, Takahara T, Nitta H, Otsuka K, Koeda K. World J Surg Oncol. 2020 Jul 16;18(1):170. doi: 10.1186/s12957-020-01955-z. PMID: 32677964

9.      Spade-Shaped Anastomosis Following a Proximal Gastrectomy Using a Double Suture to Fix the Posterior Esophageal Wall to the Anterior Gastric Wall (SPADE Operation): Case-Control Study of Early Outcomes. Han WH, Eom BW, Yoon HM, Ryu J, Kim YW. J Gastric Cancer. 2020 Mar;20(1):72-80. doi: 10.5230/jgc.2020.20.e5. Epub 2020 Feb 17. PMID: 32269846

A potential reader would observe that these studies were not included in the analysis. What was the rationale behind that? Studies only published in English were included? Studies with T1N0M0 subgroups with insufficient data of interest were excluded?

Similarly, primary data of interest along with clear definitions regarding patient selection (cancer staging eg T1N0M0 only or T1NxM0 included?), types of operative techniques (eg is pylorus preserving gastrectomy included in the LsTG group?) and types of LN stations included in each resection are expected in detail so that the reader can critically appraise the outcomes and potentially reproduce the techniques.

In the same way, a lack of a structured Results section leads to analysis’ inconsistencies making it difficult to compare the techniques (Table 5), to appropriately interpreter the outcome data and thus justify the conclusions in the Conclusions section.

In summary:

The manuscript’s analysis and therefore results are compromised due to possible selection biases and are not reproducible, as clear definitions are not given in the methods section. The tables show only part of the existing data, which makes difficult for the reader to critically appraise the results/outcomes in the conclusion. A concise structure with Methods- Study Population- Results supported by the relevant tables with appropriate technical analysis of the data would make the discussion more comprehensive for the reader and the outcomes/conclusions justified by more accurate/clear evidence throughout the manuscript.

Reviewer 2 Report

Dear Authors, thank you for allowing me to review your interesting paper dealing with function preserving gastrectomies for proximal gastric cancers. I believe it is worth publication but I would be grateful if you could specify strengths and limitations of your paper. Furthermore, the vast majority of your references are papers from Asia and this may be considered a selection bias. Please discuss.

Reviewer 3 Report

I enjoyed reading this nice review on the status of reconstruction in laparoscopic proximal gastrectomy (LPG) and laparoscopic distal gastrectomy with a small remnant stomach (LsTG).

I have the following comments/questions:

1. In the 'Reconstruction methods following LPG' Section, "anti-reflux devices" is mentioned. I would use "techniques" instead.

2. In 'Indication of LsTG' Section, "...or in the fundus is out of indication for this procedure" may be better written : "...or in the fundus is not an  indication for this procedure" if that is what the authors meant.

3. The authors summarize oncologic outcomes but fail to describe lymph node harvest. Please address for both techniques.

4. Indications for surgery as described was cT1N0. Can the authors address upstaging?
